# The Effect of Ni Interlayer on the Hot-Rolled and Quenched Stainless Steel Clad Plate

**DOI:** 10.3390/ma13235455

**Published:** 2020-11-30

**Authors:** Zengmeng Lin, Shuai Wang, Jun He, Baoxi Liu, Cuixin Chen, Jianhang Feng, Xin Zhang, Wei Fang, Fuxing Yin

**Affiliations:** 1TianJin Key Laboratory of Materials Laminating Fabrication and Interfacial Controlling Technology, Research Institute for Energy Equipment Materials, School of Materials Science and Engineering, Hebei University of Technology, Tianjin 300132, China; Lin_Zengmeng@163.com (Z.L.); karenccx@126.com (C.C.); 13933765819@126.com (J.F.); zhang_xin@hebut.edu.cn (X.Z.); rieema_fangwei@163.com (W.F.); yinfuxing@hebut.edu.cn (F.Y.); 2School of Materials Science and Engineering, Harbin Institute of Technology, P.O. Box 433, Harbin 150001, China; wangshuai190621@126.com

**Keywords:** stainless steel clad plate, vacuum hot-rolling, Ni interlayer, quenching temperature, mechanical properties

## Abstract

The vacuum hot-rolled SUS314/Q235 stainless steel clad plate has many drawbacks including serious interface alloy element diffusion, stainless steel cladding’s sensitization, and carbon steel substrate’s low strength. In this study, the comprehensive properties were systematically adjusted by changing the thickness of the Ni interlayer (0, 100, 200 μm) and the quenching temperature (1000~1150 °C). The results showed that the Ni interlayer can obviously hinder the diffusion of carbon element, so as to achieve the purpose of eliminating the decarburized layer and reducing the carbon content of the carburized layer. Meanwhile, the perfect metallurgical bonding between the substrate and cladding can be obtained, effectively improving the stainless steel clad plate’s tensile shear strength and comprehensive mechanical properties, and significantly reduce the brittleness of the carburized layer. As the quenching temperature increases, the grains coarsening of carbon steel and stainless steel became more and more serious, and the sensitization phenomenon and the thickness of the carburized layer are gradually decreased. The stainless steel clad plate (Ni layer thickness of 100 μm) quenched at 1050 °C had the best comprehensive mechanical properties. Herein, the interface shear strength, tensile strength and the fracture elongation reached 360.5 MPa, 867 MPa and 16.10%, respectively, achieving strengthening and toughening aim. This is attributed to the disappearance of the sensitization phenomenon, the grain refinement and the lower interface residual stress.

## 1. Introduction

Stainless steel clad plates have been widely used in petrochemical industry, pipeline transportation and other fields due to its excellent mechanical properties, corrosion resistance and low economic cost [1,2,3,4,5,6]. There are mainly three fabrication methods of stainless steel clad plates: Overlaying, explosive welding, and vacuum hot rolling. Herein, the vacuum hot-rolling process has become the most widely-used method for the industrialization preparation and production of stainless steel clad plate, because it has high efficiency and low pollution to the environment [7,8,9].

The vacuum hot rolling method is to make the stainless steel cladding and the carbon steel layer experience severe plastic deformation under high vacuum degree, high rolling temperature, and rolling pressure, which promotes obvious atomic bonding and element diffusion behavior at the clad interface, thus forming good metallurgical bonding [10,11,12]. However, a decarburized layer and a carburized layer are formed on both sides of the stainless steel clad plate due to the severe diffusion of carbon elements. The decarburized layer only has ferrite due to the loss of carbon element, which decreases the hardness and strength of the decarburized layer. Meanwhile, the Cr element in the stainless steel cladding is easy to react with the C element and precipitate at the grain boundaries and grain interior, further destroying the mechanical properties and corrosion resistance of the entire cladding [11,13,14,15]. Mas et al. [16] reported that C and Cr elements are prone to react to form Cr_23_C_6_ particles due to the slow cooling rate after welding, which consumes a large amount of Cr elements, and thus reducing the corrosion resistance. Pommier et al. [17,18] found that Cr_23_C_6_ particles are enriched in the grain boundary, resulting in local chromium depletion near the grain boundary, which makes it easy to form the severe intergranular corrosion crack. Chen et al. [19] reported that the addition of Ni interlayer can be used as the diffusion barrier to prevent the intergranular corrosion of the stainless steel cladding. Wang et al. [20] found that the clad plate (SUS316/Q235) fabricated with Ni interlayer possesses the high shear strength, tensile strength, and tensile fracture elongation, which are 331 MPa, 567 MPa, and 49.63%, respectively. These mechanical properties are high than clad plate without interlayer and Nb or Fe interlayer. Liu et al. [21] reported that the intergranular-like corrosion cracks at the carburized layer can be effectively inhibited with the quenching temperature of 1050 °C. However, the real bonding state of the clad plate interface with the addition of Ni interlayer and the strengthening mechanism after heat treatment have not been systematically investigated.

In fact, the Ni interlayer can inhibit the serious diffusion behavior of alloying elements as well as eliminate the formation of brittle Cr_23_C_6_ particles and intergranular corrosion cracks [22]. In addition, the heat treatment of the stainless steel clad plate with the Ni interlayer can eliminate the decarburized layer, thereby harmonizing the strength and hardness on both sides of the clad interface, and bringing about the improvement of comprehensive performance. Therefore, the Ni interlayer and the heat treatment parameters are important factors to affect the performance of the stainless steel clad plate, which has guidance for optimizing the interface structure and improving the mechanical properties.

## 2. Experimental Procedures

Stainless steel clad plate was prepared by vacuum hot rolling with Q235 steel as a substrate and 314 stainless steel as a cladding. It has been revealed in the previous work that clad plate with vacuum degree of 10^−2^ Pa obtains less interface oxides and better overall mechanical properties. Therefore, we prepared stainless steel clad plate at 10^−2^ Pa in this work [23]. In addition, two sets of stainless steel cladding plates were prepared, that is, metal Ni plates as an interlayer with a thickness of 100 μm and 200 μm were respectively added in the middle, as shown in Figure 1a. Table 1 shows the chemical composition of Q235 steel and 314 stainless steel.

The size of Q235 steel is 200 × 200 × 60 mm, and the size of 314 stainless steel is 200 × 200 × 12 mm. The surface of the sample is cleaned by an angle grinder and sandpaper to remove the oxide layer and stains. A symmetrical sequence of Q235/SUS314/isolating cloth/SUS314/Q235 was placed in a carbon steel square box, which is helpful to prevent tilting and improving fabrication efficiency. Then, it is sealed into an iron billet box by welding, and a hole is left on one side. Finally, sealed it when the vacuum reached 10^−2^ Pa. The other two groups were prepared by adding Ni interlayer between Q235 and SUS314 in the same way. The box was placed in a 1200 °C heating furnace for 120 min, and then a hot rolling experiment was performed. After 8 rolling passes, the reduction ratio reached to 90%, and finally the samples were cooled in air.

In the experiment, clad plates were divided into 3 groups by different thicknesses (0, 100, 200 μm) of Ni interlayer. Then the stainless steel clad plate of each group was cut to obtain five small plates, four of the above samples were performed under four different heat treatments of 1000 °C, 1050 °C, 1100 °C and 1150 °C, and holding for 4 min. At last, the quenching treatment was performed, and a total of 15 clad plates were obtained.

The microstructure and morphology of stainless steel clad plate were observed by optical microscope (OM, Leica, Veclar, Germany). The solutions of 10% CrO_3_ + 90% H_2_O and 4% nitric acid ethanol were used as corrosive agents to show the microstructures of stainless steel cladding and carbon steel substrate, respectively. In addition, depth of field optical microscope (Zeiss, Oberkochen, Germany) JSM-7100F scanning electron microscope (SEM, JEOL, Tokyo, Japan), Tecnai G2 F30 transmission electron microscope (TEM, FEI, Portland, ORE, USA) with energy dispersive spectrometer (EDS) and JXA-8530F electron probe microanalysis (EPMA, JEOL., Tokyo, Japan) were used in the experiment to investigate interface microstructure. As shown in Figure 1a, it is a schematic diagram of the cross section of the sample before rolling, where the “carbon steel” represents the outermost blank box. Strength properties were measured at a constant speed of 2 mm/min using an AGS-50kNX universal testing machine (Shimadzu, Kyoto, Japan). As shown in Figure 1b, the width of all shear samples is 5.0 mm, and the gap distance is 1.5 mm. Figure 1c shows an asymmetric tensile sample with a size of 18 × 2.4 × 2.4 mm. The carbon steel substrate and the stainless steel layer in the middle region are both 1.2 mm. Herein, the tensile and shear samples were ground and polished before the testing. During the HV hardness testing (Shimadzu, Kyoto, Japan), a 100gf test load was applied for all the samples. Taking the clad interface as the origin, the hardness test is performed at intervals of 10 μm on both sides. The hardness value is taken as average of five times that performed at each interval (21 points in total).

## 3. Results and Discussion

Figure 2 shows the microstructure of the stainless steel clad plate with different thickness of Ni interlayer and different quenching temperatures. As shown in Figure 2a, excessive diffusion of the carbon element would form carburized and decarburized layers on both sides of the clad interface without Ni interlayer. With the addition of Ni interlayer, as shown in Figure 2b,c the Ni interlayer still maintains a straight, uniform and continuous shape after hot rolling, which indicates that the Ni interlayer and the carbon steel base layer/stainless steel cladding layer had good deformation coordination and uniform plastic deformation ability at high temperature, avoiding the occurrence of interface defects such as local necking or fracture. At the same time, the straight and uniform Ni interlayer is beneficial to inhibit the excessive diffusion of alloying elements. It will effectively eliminate the decarburized layer and reduce the carbon content of the carburized layer. However, because of the high carbon content of SUS314 stainless steel and the slow cooling process after hot rolling, it is inevitable that sensitization occurs at 450–820 °C. Cr_23_C_6_ is easy to precipitate at grain boundaries, resulting in a decrease in corrosion resistance. Therefore, it can be observed in Figure 2a–c that the stainless steel cladding is easily corroded and exhibits obvious intergranular corrosion. In order to eliminate the serious sensitization phenomenon of the SUS314/Q235 stainless steel clad plate, the three materials were quenched at different temperatures, as shown in Figure 2d–o. Obviously, the sensitization phenomenon of the stainless steel cladding that far from the interface is basically eliminated, while there is still a carburized layer near the interface. Interestingly, as the quenching temperature increases, the thickness of the carburized layer gradually decreases, from 172 μm to 38 μm. At the same time, it was found that the addition of Ni interlayer did not significantly change the thickness of the carburized layer. Moreover, after quenching at 1150 °C, although the carburized layer has basically disappeared, the stainless steel cladding has obvious abnormal grain coarsening, which will seriously affect the overall mechanical properties. The original decarburized layer still contains a small amount of carbon element [22], and the mixed structure of ferrite and martensite will be formed after quenching, which will significantly enhance the shear and tensile strength of the base layer. However, it can be observed from the above metallographic images that the thickness of the Ni interlayer did not significantly change the microstructure of the stainless steel clad plate.

Figure 3 displays the EPMA distribution of alloying elements at the interface transition zone of hot-rolled stainless steel clad plates with different thicknesses of Ni interlayers. Obviously, the O element distribution diagrams in Figure 3a shows that a large number of dot-like oxide inclusions are distributed at the clad interface without adding Ni interlayer, showing a uniform dispersed distribution. Through the comparison of Fe, Cr, and Mn elements, it can be found that the oxide inclusions are mainly existed in the form of Cr-Mn oxides, which is consistent with the MnCr_2_O_4_ reported by Xie et al. [24]. However, as shown in Figure 3b, with the addition of the 100 μm Ni interlayer, the oxide near the clad interface decreased significantly. This is because the addition of the Ni interlayer would increase an interlayer interface area which leads to the reduction of interfacial oxide per unit area under the premise that the total oxygen content remains unchanged. Most interestingly, the interface oxide is mainly distributed along the clad interface between the stainless steel and the Ni interlayer, which coincides with the tensile–shear fracture position of the subsequent quenched stainless steel clad plate, indicating that there is a weak area of interface of stainless steel clad plate. After hot rolling, the oxide at the interface always remains in a stable form, finally formed oxides with Cr, Mn, and Si elements, and almost no Fe oxide [24]. Sun et al. [25] reported that under high temperature, high vacuum, and high pressure, there will be initial Fe oxides at the interface, which will be gradually replaced by stable oxides of Cr, Mn, Si, and Al through the selective oxidation. Therefore, under high rolling reduction, there is no obvious oxide at the clad interface between the carbon steel and Ni interlayer while there are mainly a large amount of small dot-like oxides such as MnCr_2_O_4_ and MnSi_2_O_4_ were located at the clad interface between the stainless steel and Ni interlayer, which will affect the bonding performance between the Ni interlayer and the stainless steel cladding. In addition, as shown in Figure 3c, the oxide content at the interface of the stainless steel clad plate with the thickness of 200 μm Ni interlayer is basically the same as that of the 100 μm Ni interlayer. It shows that under the premise of the same interface area, the thickness of the interlayer has almost no effect on the interface oxide. As a whole, the addition of the Ni interlayer can avoid the excessive oxidation of the stainless steel clad plate, thereby effectively improving the interface bonding state.

Figure 4 shows the TEM micrograph of the clad interface with the addition of 100 μm Ni interlayer. There is no obvious interlayer interface between the Ni interlayer and the stainless steel cladding/the carbon steel layer. As shown in Figure 4a,b,e–h, the approximate interface between the Ni interlayer and the substrate can be found from the element distribution on both sides. The obvious Fe peak can be found in Figure 4e, combined with Table 2, the content of Fe is 81.39 at.%, indicating that the layer is already close to the carbon steel side. The high content of Ni in points F and G indicates that these two areas are the location of the Ni interlayer. The point H is located in the stainless steel cladding, because the content of Cr element reaches 12.15 at.%, which indicates that the area is already located at the stainless steel layer. As shown in Figure 4e-g, the Fe element diffuses in the Ni interlayer and presents a gradient distribution state, which means that the significant diffusion phenomenon has occurred between the Ni interlayer and the substrate. Combined with Figure 4a,b it can be observed that many grains across the interface transition zone. It indicates that obvious metallurgical bonding has occurred at the clad interface, which will greatly improve the interface bonding strength [20].

In addition, as shown in Figure 4c, lamellar pearlite appears near the interface, and ferrite also appears around the pearlite. The ferrite and pearlite mixture microstructure are same as that of the matrix (Q235), which proves that there is absence of decarburized layer. This is attributed to the fact that the Ni interlayer can significantly inhibit the diffusion of C elements. In addition, in Figure 4d, there are many chromium carbides (Cr_23_C_6_) particles near the interface on the stainless steel side, which indicates that the hot-rolled stainless steel cladding exhibits obvious sensitization, leading to the tendency of intergranular corrosion.

Figure 5 shows the microstructure diagram of the interface transition zone of a stainless steel clad plate with the addition of 100 μm Ni interlayer and quenched at 1050 °C. As shown in Figure 5a,b,f–h similar to the hot-rolled interface, there is no obvious clad interface between the Ni interlayer and the stainless steel/carbon steel. Moreover, there are traces of Fe diffusion in Ni interlayer as shown in Table 3, which proves that the existence of the Ni interlayer can effectively improve the interface bonding state. In addition, Figure 5c shows the martensite structure is formed after quenching at 1050 °C. Therefore, a large amount of martensite was formed in the carbon steel near the interface due to the disappearance of the decarburized layer, thereby increasing the strength and hardness. In addition, Figure 5d shows the austenite stainless steel cladding area far from the interface contains many twins, and there is no obvious precipitated phase near the grain boundaries, indicating that the heat treatment has significant solid solution effect on eliminating the sensitization phenomenon. In addition, Figure 5e shows the area of the stainless steel layer close to the interface where there are a large number of Cr_23_C_6_ particles with a size of 220 nm can be seen at the grain boundary, which indicates that there is still a certain thickness of carburized layer near the interface, and is also consistent with the metallographic morphology in Figure 2. In summary, the quenching can obviously eliminate the sensitization phenomenon of the stainless steel cladding, while quenched at 1050 °C cannot completely eliminate the carburized layer due to the higher carbon content near the carburized layer.

Figure 6 shows the hardness distribution of hot-rolled stainless steel clad plates with different Ni interlayer thicknesses. As shown in Figure 6a, stainless steel clad plate without Ni interlayer presents different indentation sizes in various regions due to diffusion behavior of C element, and the sequences of indentation sizes are as follows: Carburized layer < stainless steel cladding < carbon steel substrate < decarburized layer, which is in agreement with HV-distance curves in Figure 6d [26]. In order to ensure the accuracy of the experimental results and ensure a reasonable distance between the indentations. In the direction perpendicular to the interface, select an area for hardness test at intervals of 10 μm. In this area, perform 5 tests along the parallel interface direction at intervals of 50 μm, and take the average value. In the experiment, a part of the test points will appear in different positions consistent with the distance from the interface, not in the area shown in Figure 6a–c. Herein, the highest hardness value of 344 HV corresponds to smallest indentation of carburized layer, while lowest hardness value of 120 HV corresponds to largest indentation of decarburized layer. It is shown that excessive diffusion of carbon will cause a substantial increase in the local hardness of the carburized layer and a significant decrease in the local hardness of the decarburized layer, resulting in the occurrence of tunnel cracks in the carburized layer and a decrease in the bearing capacity of the decarburized layer. However, after adding the 100 μm Ni interlayer, the difference of indentation sizes in different regions is obviously decreased, as shown in Figure 6b. Moreover, the decarburized layer is disappeared, and the hardness of carburized layer decreases. It can improve the weak area’s strength and crack propagation resistance of the clad plate. As shown in Figure 6d, with the addition of a 100 μm Ni interlayer, the peak hardness values of the carburized layer decreased from 344 HV to 269 HV, and the hardness of the decarburized layer increased from 120 HV to 142 HV. The decreased decarburized layer and the hardness difference of the carbon layer effectively reduces the interface residual stress and improves the overall coordinated deformation ability. In addition, the indentation size and hardness value of the Ni interlayer fall between the carbon steel layer and the carburized layer, which effectively buffers the local stress difference of the interface transition layer and improves the uniform plastic deformation ability of the stainless steel clad plate. Interestingly, the indentation size and hardness behavior of the stainless steel clad plate with a 200 μm Ni interlayer is basically similar to that of the 100 μm Ni interlayer. Although the hardness value of the carburized layer is further reduced to 252 HV, and the carbon steel side’s hardness value peak reached 165 HV. It shows that increasing the thickness of the Ni interlayer only slightly inhibit the diffusion of carbon elements. In summary, the addition of the Ni interlayer can significantly reduce the carbon diffusion behavior. It can avoid the appearance of the decarburized layer, and effectively reduce the carbon content at the carburized layer.

Figure 7 shows the tensile shear stress–displacement curves of stainless steel clad plates with different Ni interlayer thicknesses and different quenching temperatures, and the shear strengths are listed in Table 4. All the shear strengths of the stainless steel clad plates were higher than the international standard value of 210 MPa. The shear strengths of the hot-rolled stainless steel clad plates with the Ni interlayers of 0, 100, and 200 um were 360, 393, and 387 MPa, respectively. It is indicated that the shear strength had been significantly improved with the addition of the Ni interlayer, which is due to the inhibition of C element’s diffusion and then eliminates the decarburization layer. Meanwhile, the thickness of the Ni interlayer has no significant effect on its shear strength. It is worth noting that with the increase of the quenching temperature, the shear strength of the stainless steel clad plate increases first and then decreases. All shear strength of the clad plates reached the maximum value at 1050 °C, which was 270.1, 360.5, and 345 MPa, respectively. That is, as the severe diffusion of elements at high temperature will obviously improve the interface bonding strength. However, the increased of the interface residual stress and the coarsening of the structure will deteriorate the interface bonding strength, so that the tensile–shear strength exhibited peak characteristics. This was generally lower than the shear strength of the hot rolled state, it was due to the obvious residual internal stress on both sides of the interface during the quenching and then deteriorates the interface bonding strength. In summary, 100 μm Ni interlayer and quenching at 1050 °C can eliminate the decarburized layer and ensure higher shear strength. Herein, the shear displacement was 2.099 mm, and the shear strength was 360.5 MPa.

Figure 8 shows the profile tensile shear fracture characteristics of stainless steel clad plates with different Ni interlayer thicknesses and different quenching temperatures. In Figure 8a–c, it can be clearly observed that the fractured parts appear in the decarburized layer and the carbon steel layer away from the interface, and the shear fracture had obvious warping behavior. The decarburized layer has lower strength and hardness than interface, which became the weak part of the stainless steel clad plate. Therefore, although the gap appeared at the interlayer interface, the decarburized layer and carbon steel were still crack resource. It was the reason that the shear fracture was broken at decarburized layer, and also explained that the real interface bonding strength was higher than the measured shear strength. The thickness of the Ni interlayer had no obvious effect on the microstructure of the carbon steel layer near the interface, so the tensile shear strength was basically the same. It was very interesting that, compared with the hot rolled state, the shear fractures of the quenched stainless steel clad plate were located at the interface between the stainless steel cladding and the Ni interlayer, and the wrapping was not very obvious. Compared to the carbon steel and Ni interlayer, there were many oxides at the interface between the stainless steel cladding and the Ni interlayer, and the strength of the carbon steel layer was greatly increased after quenching. Therefore, the interface between the Ni interlayer and the stainless steel cladding becomes the weakest part, which became the initiation area of shear cracks. In addition, the difference in thermal expansion coefficients of dissimilar metals and the phase transitions would generate severe residual stresses at interface during the quenching temperature, so that greatly reduced the shear strength of the interface. Therefore, the tensile shear strength of the heat-treated state was significantly lower than that of hot rolled state.

Figure 9 shows the tensile engineering stress-strain curves of stainless steel clad plates with different interlayer thickness and quenching temperature, and the corresponding tensile properties were listed in Table 5. The tensile strength of the stainless steel clad plate had been significantly improved with the addition of 100 μm Ni interlayer, while the fracture elongation is basically the same. The addition of the Ni interlayer eliminates the decarburized layer and reduces the carbon element in the carburized layer, so that increased the overall strength of the carbon steel layer and avoided the tendency of intergranular fracture of the stainless steel cladding. It would effectively realize the strengthening–toughening aim of the stainless steel clad plate. However, when the thickness of the Ni interlayer reaches 200 μm, the tensile strength and fracture elongation of the stainless steel clad plate did not change significantly, indicating that the thickness of Ni interlayer would not significantly change the interface bonding performance of the stainless steel clad plate. In addition, the tensile strength of the stainless steel clad plate was greatly improved by quenching treatment. As the quenching temperature increases, the tensile strength of the stainless steel clad plate first increases and then decreases. Because of the structure on the carbon steel side was transformed into martensite or a mixed structure of martensite and ferrite after quenching, thereby increasing the strength of the carbon steel side. More importantly, the sensitization phenomenon of the stainless steel cladding was eliminated. Meanwhile, the intergranular chromium carbide of the carburized layer would be solid-solved at high temperatures. It slows down the generation of intergranular corrosion cracks and effectively improving the plastic deformation capacity of stainless steel cladding. However, as the temperature rises further, the tensile strength of the stainless steel cladding plate begin to decreased, which is caused by the coarsening of the structure of the stainless steel clad plate. All in all, with the addition of 100 μm Ni interlayer and quenching at 1050 °C, the stainless steel clad plate had the best tensile properties. The yield strength, tensile strength and elongation after fracture were 520 MPa, 867 MPa, and 16.10%, respectively.

Figure 10 shows the tensile fracture characteristics of stainless steel clad plates with different thickness of Ni interlayer and quenching temperature. As shown in Figure 10a,f, the hot-rolled stainless steel clad plate did not undergo delamination, indicating that hot-rolled stainless steel clad plates have strong interface. The tensile fracture morphology of the quenched state shows that with the increase of the quenching temperature, different degrees of delamination occurred. Among them, the delamination at 1050 °C was the slightest, and the delamination at 1150 °C was the most serious. It was also consistent with the tensile shear performance. At high temperatures, the diffusion of elements at the interface will obviously increase the bonding strength of the interface. Meanwhile, the coarsening of the microstructure and the increase of the residual stress of the interface after quenching would deteriorate the interface bonding strength. The competitive relationship between these three makes the interface bonding strength of the stainless steel clad plate presents the unique characteristics of first increasing and then decreasing trend. After quenching at 1050 °C, the stainless steel cladding, carbon steel layer and Ni interlayer had good deformation coordination due to the high interlayer bonding performance, then the highest fracture elongation after fracture was obtained. From the overall point of view, the fracture surface had obvious necking phenomenon, which was a typical microporous aggregation fracture mode. It shows that the clad plate had superior plastic deformation ability.

To sum up, in order to vividly clarify the influence of the addition of Ni interlayer and subsequent quenching treatment on the microstructure of stainless steel clad plate, Figure 11 shows a schematic diagram of the microstructure evolution of stainless steel clad plates with different thicknesses of Ni interlayers and different quenching temperatures. There were obvious carburized and decarburized layers on both sides of the interface due to the excessive diffusion of carbon during the hot rolling process as shown in Figure 11a. The stainless steel cladding had obvious sensitization due to the slow cooling rate, and there was many dispersed Cr_23_C_6_ at the grain boundary, and the carbon steel matrix was a mixed structure of ferrite and pearlite. Figure 11b,c are the hot-rolled structures of stainless steel clad plates with the addition of 100 μm and 200 μm Ni interlayers. Because of the Ni interlayer had a significant inhibitory effect on the C element, the decarburized layer disappears and the C element content of carburized layer was greatly reduced, while the carbon steel layer and stainless steel cladding structure had not changed significantly. At the same time, there were symbiotic grains and obvious element diffusion between the substrate and the Ni interlayer, and then forming a strong metallurgical interface. Figure 11d–f is the microstructure of stainless steel clad plate with Ni interlayer at different quenching temperatures. Obviously, there is no sensitization phenomenon in the stainless steel layer that far from the interface, and the carbon steel layer had become a martensite structure. Moreover, as the temperature increases, the thickness of the carburized layer was significantly reduced. The stainless steel cladding had abnormal growth of crystal grains, leading to the decrease of comprehensive mechanical properties.

## 4. Conclusions

In this work, we have achieved the triple purpose of eliminating the serious carbon diffusion and sensitization phenomenon of hot-rolled stainless steel clad plate, as well as effectively improving the bearing capacity by adding Ni interlayer and subsequently quenching treatment. The achieved results are as follows:The addition of the Ni interlayer can obviously hinder the diffusion of carbon element, effectively reduce the thickness of the decarburized layer and carbon content of the carburized layer. Many grains across the interface transition zone, which significantly improves the interface bonding strength. Quenching treatment can obviously eliminate the sensitization phenomenon of the stainless steel cladding, and the thickness of the carburized layer gradually decreased with the increase of quenching temperature.Regardless of hot-rolled or heat-treated state, the shear strength, tensile strength and fracture elongation of stainless steel clad plates with the addition of the Ni interlayer were higher than those without the interlayer, and the stainless steel clad plate with 100 μm Ni interlayer had the highest comprehensive mechanical properties. This is related to its excellent interfacial metallurgical bonding, the disappearance of the decarburized layer, the reduction of the carbon content of the carburized layer, and the relatively low proportion of Ni interlayer.For the hot-rolled stainless steel clad plate, due to the lower bearing capacity of the weak part, the shear fracture zone was located in the decarburized layer, while after the Ni layer is added, the fracture zone was located at the carbon steel side, which indicates that the actual interface shear strength will be higher than experimental measured values. After quenching, due to the ultra-high hardness and strength of the martensite matrix at the carbon steel, the fracture zone was located at the interface between the stainless steel and the carbon steel/Ni interlayer.Regardless of whether the interlayer is added or not, the heat-treated stainless steel clad plate that quenched at 1050 °C had the highest tensile shear strength, tensile strength and fracture elongation. This is due to the disappearance of the sensitization phenomenon, the grain refinement and the lower residual stress at the interface.

## Figures and Tables

**Figure 1 materials-13-05455-f001:**
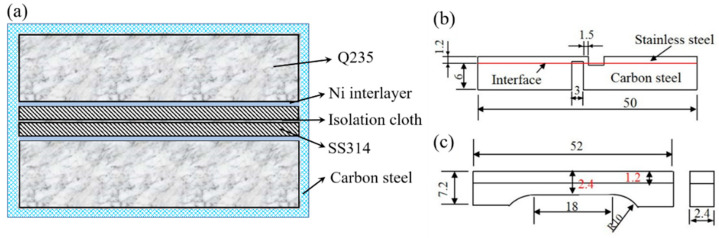
Schematic of specimen shape under testing. (**a**) schematic illustration of stacking build-up slab; (**b**) tensile–shear specimen; (**c**) non-symmetrical tensile specimen.

**Figure 2 materials-13-05455-f002:**
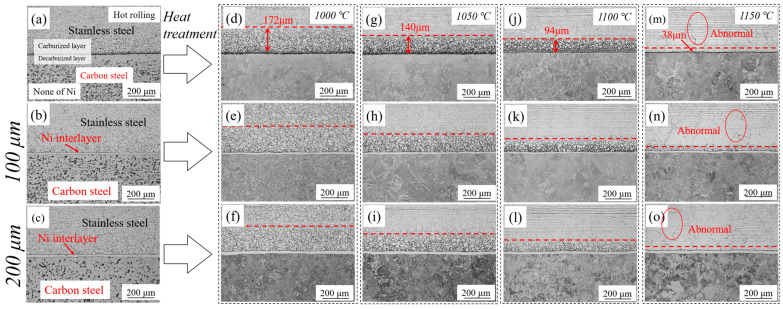
The optical microstructure of stainless steel clad plate with Ni interlayers with the thickness of 0, 100, and 200 μm and quenching temperatures at (**a**–**c**) hot rolling; (**d**–**f**) 1000 °C; (**g**–**i**) 1050 °C; (**j**–**l**) 1100 °C; (**m**–**o**) 1150 °C.

**Figure 3 materials-13-05455-f003:**
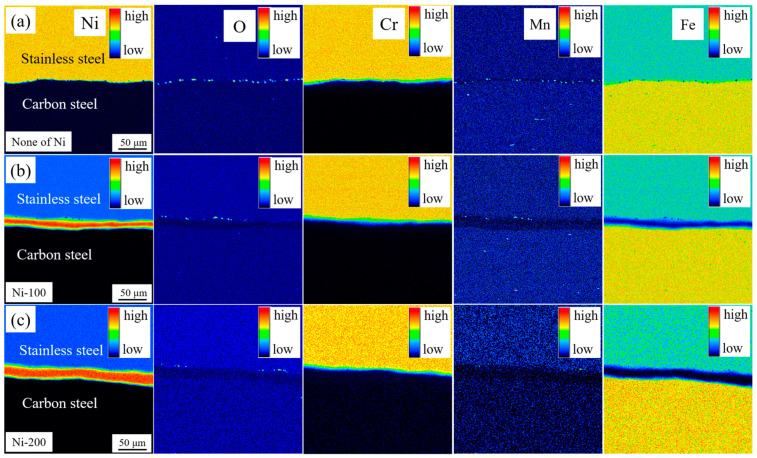
The electron probe microanalysis (EPMA) maps of Ni, O, Cr, Mn, Fe elements distribution at clad interfaces with different thickness Ni interlayer of (**a**) 0 μm; (**b**) 100 μm; (**c**) 200 μm.

**Figure 4 materials-13-05455-f004:**
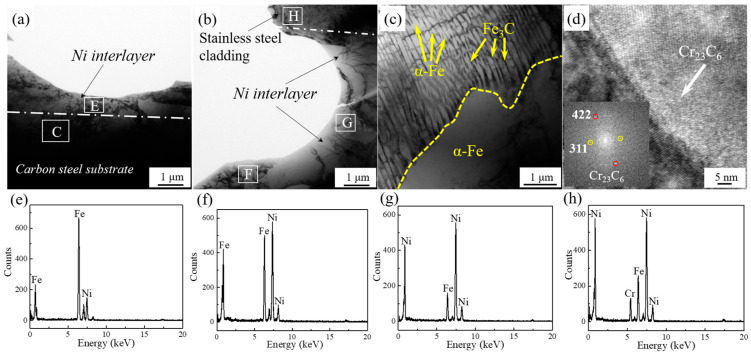
The TEM microstructure of hot rolled clad plate with 100 μm Ni interlayer at interface zone. (**a**) Ni-Carbon steel; (**b**) Ni-Stainless steel; (**c**) Area C indicated by the rectangle in (**a**); (**d**) Stainless steel; (**e**) EDS results of area E indicated by the rectangle in (**a**); (**f**–**h**) EDS results of areas F, G and H indicated by the rectangle in (**b**).

**Figure 5 materials-13-05455-f005:**
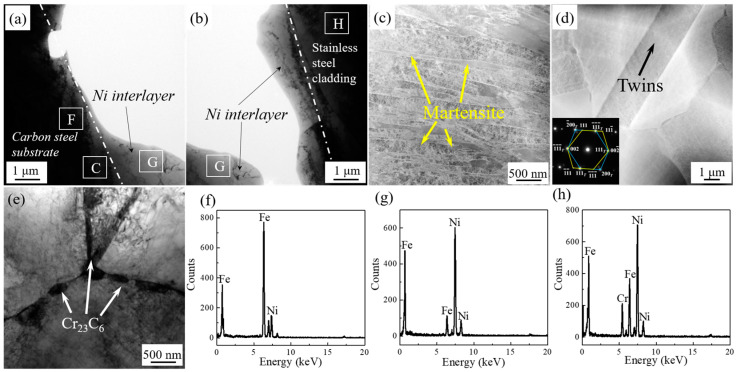
The TEM microstructure of 1050 °C heat treatment clad plate with 100 μm Ni interlayer at interface zone. (**a**) Ni-Carbon steel; (**b**) Ni-Stainless steel; (**c**) Area C indicated by the rectangle in (**a**); (**d**) Stainless steel. The lower left inset is selected area electron diffraction (SAED); (**e**) Stainless steel; (**f**) EDS results of area F indicated by the rectangle in (**a**); (**g**–**h**) EDS results of area G and H indicated by the rectangle in (**b**).

**Figure 6 materials-13-05455-f006:**
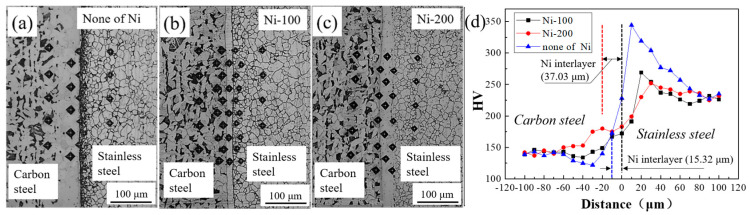
The hardness of stainless steel clad plates. (**a**–**c**) hardness indentation with different thicknesses of Ni interlayer (Partial area); (**d**) the distribution of hardness.

**Figure 7 materials-13-05455-f007:**
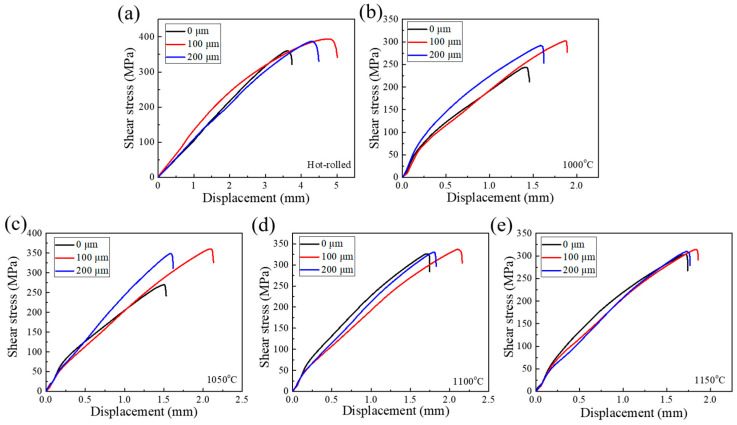
The tensile shear stress-displacement curves of stainless steel clad plate with different thicknesses of Ni interlayer (0, 100, and 200 μm), hot rolled (**a**) and quenching temperatures at (**b**) 1000 °C, (**c**) 1050 °C, (**d**) 1100 °C, and (**e**) 1150 °C.

**Figure 8 materials-13-05455-f008:**
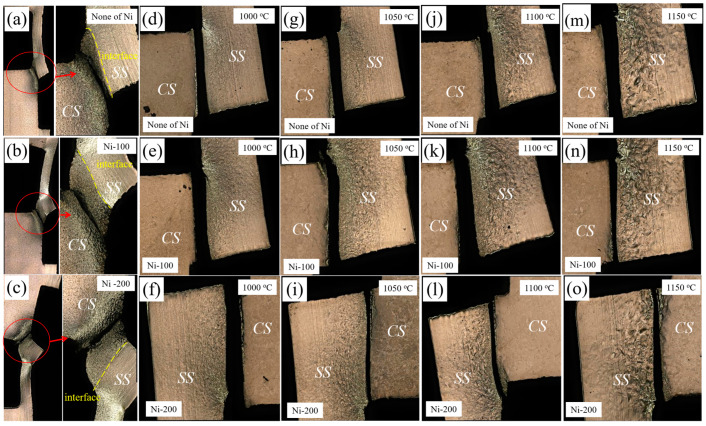
The profile tensile shear fracture characteristics of stainless steel clad plates with 0, 100, and 200 μm thicknesses of Ni interlayer and quenching temperatures at (**a**–**c**) hot rolled; (**d**–**f**) 1000 °C; (**g**–**i**) 1050 °C; (**j**–**l**) 1100 °C; and (**m**–**o**) 1150 °C. (CS—Carbon steel; SS—Stainless steel).

**Figure 9 materials-13-05455-f009:**
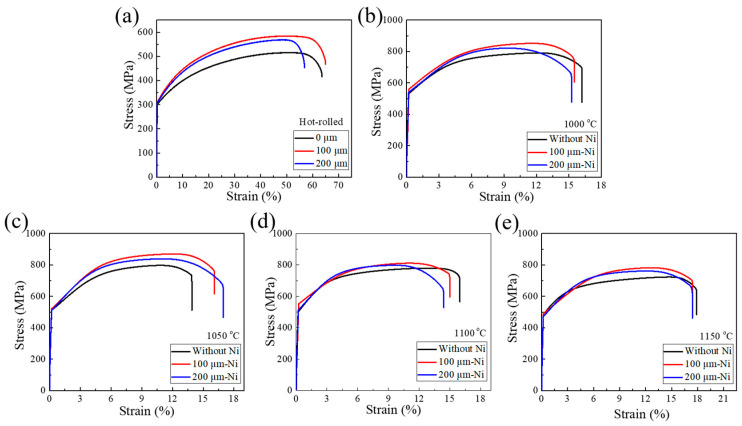
The tensile stress-strain curves of stainless steel clad plates with different thicknesses of Ni interlayer (0, 100, and 200 μm), hot rolled (**a**) and quenching temperature at (**b**) 1000 °C; (**c**) 1050 °C; (**d**) 1100 °C; and (**e**) 1150 °C.

**Figure 10 materials-13-05455-f010:**
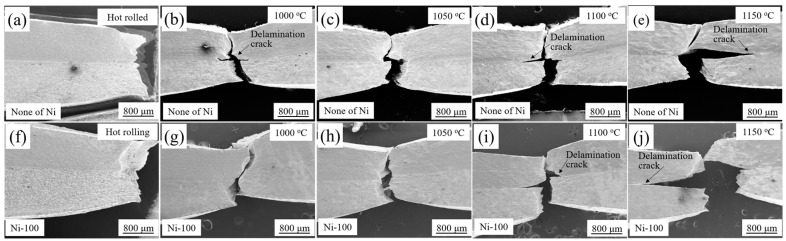
The profile tensile fracture characteristics of hot rolled stainless steel clad plates and with quenching temperatures at 1000 °C, 1050 °C, 1100 °C, and 1150 °C. (**a**–**e**) without Ni interlayer; (**f**–**j**) with Ni interlayer.

**Figure 11 materials-13-05455-f011:**
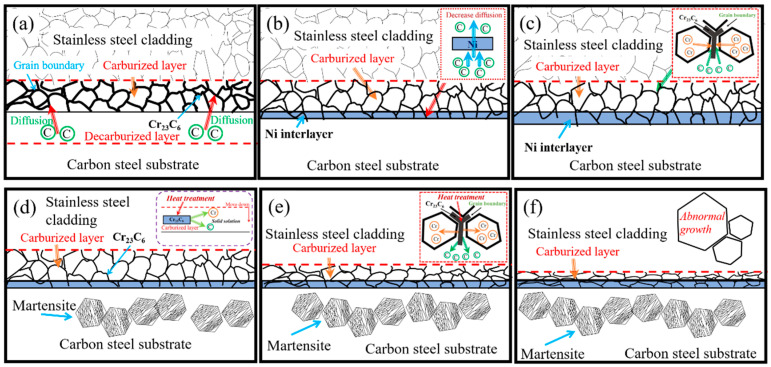
The schematic diagram of bonding interface formation process of stainless steel clad plates with different thickness of Ni interlayer and quenching temperatures. (**a**) Without Ni interlayer; (**b**) 100 μm Ni interlayer; (**c**) 200 μm Ni interlayer; (**d**) 1000 °C; (**e**) 1050 °C; and (**f**) 1150 °C.

**Table 1 materials-13-05455-t001:** The chemical compositions of the carbon steel and stainless steel (wt.%).

Elements	Fe	Cr	Ni	C	Mn	Si	P	S
Q235	Bal	-	-	0.18	0.5	0.3	0.045	0.05
SUS314	Bal	24	20	0.23	1.8	2	0.04	0.02

**Table 2 materials-13-05455-t002:** EDS results of hot-rolled clad plate at interface zone.

Position	Ni (at.%)	Fe (at.%)	Cr (at.%)
E	18.60	81.39	-
F	56.16	43.83	-
G	81.30	18.69	-
H	63.14	24.69	12.15

**Table 3 materials-13-05455-t003:** EDS results of 1050 °C heat treatment clad plate at interface zone.

Position	Ni (at.%)	Fe (at.%)	Cr (at.%)
E	16.41	83.58	-
F	87.44	12.55	-
G	58.86	26.49	14.63

**Table 4 materials-13-05455-t004:** The tensile strengths and displacements of stainless steel clad plates with different thicknesses of Ni interlayer and quenching temperatures.

Interlayer (μm)-Temperature (°C)	Shear Strength (MPa)	Shear Displacement (mm)
0-hot rolled	360	3.743
100-hot rolled	393	4.991
200-hot rolled	387	4.493
0-1000	243.5	1.425
100-1000	321.3	1.863
200-1000	291.8	1.584
0-1050	270.1	1.506
100-1050	360.5	2.099
200-1050	329.3	1.586
0-1100	326.1	1.713
100-1100	336.8	2.104
200-1100	317.3	1.791
0-1150	303.1	1.718
100-1150	314.2	1.84
200-1150	274.2	1.728

**Table 5 materials-13-05455-t005:** The tensile properties of stainless steel clad plates with different thicknesses of Ni interlayer and quenching temperatures.

Interlayer (μm)-Temperature (°C)	Yield Strength (MPa)	Ultimate Strength (MPa)	Fracture Elongation (%)
0-hot rolled	300	514	64.10
100-hot rolled	302	583	65.27
200-hot rolled	300	568	57.11
0-1000	530	790	16.20
100-1000	560	852	14.11
200-1000	540	793	15.20
0-1050	510	796	14.00
100-1050	520	867	16.10
200-1050	510	800	17.10
0-1100	500	780	17.00
100-1100	550	810	12.78
200-1100	510	798	14.40
0-1150	485	723	17.90
100-1150	490	780	17.40
200-1150	470	715	19.30

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
