# Peer review of "The Effect of Ni Interlayer on the Hot-Rolled and Quenched Stainless Steel Clad Plate"

_materials, 2020, doi:10.3390/ma13235455_

Round 1

Reviewer 1 Report

There are useful experimental results in the paper, which can be interesting for the journal readers.

Among the flows of the paper are:

  1. The amount of the text is a bit redundant. For example, the authors could not to repeat so march about inhibition of Ni interlayer on C element diffusion and about decreasing of decarburization layer at carbon steel layer”…
  2. The English level is quite poor. The authors should give the paper for proofreading to someone with good English. There are more bad English-related things in the text than I pointed below.

  1. “because of its … friendly [7-9].” I think there is some translation mistake here.
  2. “Pommier et al. [17,18] found that Cr23C6 particles are enriched in the grain boundary, resulting in local chromium depletion at the grain boundary, which makes it easy to form the severe intergranular corrosion” → “… resulting in local chromium depletion near the grain boundary…”?
  3. “Therefore, we continuously prepare stainless steel clad plate at 10-2Pa in this article [23]” The sentence is not clear. What does “continuously” mean here? And “article” → “paper” or “work” or “research” etc.
  4. “The green box was placed in…” green?
  5. “and all of holding time are 4 minutes” It is clear what you mean, but there is some language mistake.
  6. “(EPMA) will be used in the experiment to investigate interface microstructure.” → “(EPMA) was used …”?
  7. “The samples were performed at a constant speed of 2 mm/min using an AGS-50kNX universal testing machine.” → May be something like that: “strength properties were measured at a constant speed …”?
  8. “Taking the interface of the clad as the origin, and the hardness test is performed at intervals of 10μm on both sides” The “and” may be excess here.
  9. “no obvious oxide at the clad interface the carbon steel and Ni interlayer” →“no obvious oxide at the clad interface between the carbon steel and Ni interlayer”
  10. “The area corresponding to the point H represents the representative element Cr of stainless steel”
  11. “it can be observed that the grain symbiosis has occurred at the interfaces” What a strange term “grain symbiosis”! Does it really exist?
  12. “Its microstructure is same as that of the matrix (Q235), which proves that there is already near the interface” What is near the interface?
  13. “Fig. 5(d) shows the stainless steel cladding area far from the interface that many twins” → “…far from the interface where there are many twins…”?
  14. “Fig. 5(e) shows the area of the stainless steel layer close to the interface, that a large number of Cr23C6 particles” The same as in the previous item.
  15. It is not simple to understand at which sides stainless and carbon steels are at the fig 6.
  16. “In order to ensure the accuracy of the experimental results and ensure a reasonable distance between the indentations” What was done to ensure that?
  17. “and the influence of phase transitions would generated large residual stresses… and greatly reduced the shear strength”
  18. “Obviously, the stainless steel cladding that far from the interface had eliminated the sensitization phenomenon” Some language mistake.

Author Response

Response to Reviewer Comments

Point 1: The amount of the text is a bit redundant. For example, the authors could not to repeat so much about inhibition of Ni interlayer on C element diffusion and about decreasing of decarburization layer at carbon steel layer”…

Response 1: Thanks for the reviewer’s suggestion, we have refined and modified the repeated part.

Page 4, line 20: “The microstructure of the carbon steel substrate changed from the mixed structure of hot-rolled ferrite and pearlite to martensite after quenching.” → delete.

Page 9, line 9: “However, after adding the 100μm Ni interlayer, the difference of indentation sizes in different regions gradually decreases as shown in Fig. 6(b), which results from the inhibition of Ni interlayer on C element diffusion. Therefore, there is no obvious decarburization layer at carbon steel layer, and the hardness of carburized layer decreases, improving the weak area’s strength and crack propagation resistance of the whole clad plate.” → “However, after adding the 100μm Ni interlayer, the difference of indentation sizes in different regions is obviously decreased, as shown in Fig. 6(b). Moreover, the decarburized layer is disappeared, and the hardness of carburized layer decreases. It can improve the weak area’s strength and crack propagation resistance of the clad plate.”

Point 2: The English level is quite poor.

Response 2: Thanks for the reviewer’s suggestion, we have carefully revised the grammar, the specific changes are as follows:

(1) Page 1, line 11: “The vacuum hot-rolled SS314/Q235 stainless steel clad plate has serious interface alloy element diffusion, stainless steel cladding’s sensitization and carbon steel substrate’s low strength. In this study, the comprehensive properties was improved by systematically controlling the thickness of the Ni interlayer (0μm, 100μm, 200μm) and the quenching temperature (1000oC~1150oC).” → “The vacuum hot-rolled SS314/Q235 stainless steel clad plate has many drawbacks including serious interface alloy element diffusion, stainless steel cladding’s sensitization and carbon steel substrate’s low strength. In this study, the comprehensive properties were systematically adjusted by changing the thickness of the Ni interlayer (0μm, 100μm, 200μm) and the quenching temperature (1000oC~1150oC).”

(2) Page 1, line 26: “Herein, the interface shear strength, tensile strength and the elongation after fracture reached 360.5 MPa, 867 MPa and 16.10%, respectively, which achieving strengthening and toughening aim. This is attributed to disappearance of the sensitization phenomenon of the stainless steel cladding, the smaller grain size and the lower interface residual stress.” → “Herein, the interface shear strength, tensile strength and the fracture elongation reached 360.5 MPa, 867 MPa and 16.10%, respectively, achieving strengthening and toughening aim. This is attributed to the disappearance of the sensitization phenomenon, the grain refinement and the lower interface residual stress.”

(3) Page 1, line 39: “because of its … friendly [7-9].” → “because it has high efficiency and low pollution to the environment.”

(4) Page 1, line 42: “high rolling temperature”.

(5) Page 2, line 3: “the hardness and strength”.

(6) Page 2, line 10: “resulting in local chromium depletion at the grain boundary, which makes it easy to form the severe intergranular corrosion.” → “resulting in local chromium depletion near the grain boundary, which makes it easy to form the severe intergranular corrosion.”

(7) Page 2, line 11: “intergranular corrosion crack”.

(8) Page 2, line 27: “the clad plate interface” → “the clad interface”.

(9) Page 2, line 28: “process” → “parameters”.

(10) Page 2, line 35: “Therefore, we continuously prepare stainless steel clad plate at 10-2Pa in this article [23].” → “Therefore, we prepared stainless steel clad plate at 10-2Pa in this work [23].”

(11) Page 2, line 47: “The green box was placed” → “The box was placed”.

(12) Page 3, line 4: “and all of holding time are 4 minutes.” → “and holding for 4 minutes.”

(13) Page 3, line 7: “(EPMA) will be used in the experiment to investigate interface microstructure.” → “(EPMA, JEOL. Japan) were used in the experiment to investigate interface microstructure.”

(14) Page 3, line 16: “The samples were performed at a constant speed of 2 mm/min using an AGS-50kNX universal testing machine.” → “Strength properties were measured at a constant speed of 2mm/min using an AGS-50kNX universal testing machine”.

(15) Page 3, line 23: “Taking the interface of the clad as the origin, and the hardness test is performed at intervals of 10μm on both sides.” → “Taking the interface of the clad as the origin, the hardness test is performed at intervals of 10μm on both sides.”

(16) Page 5, line 23: “or be observed by the matrix”

(17) Page 5, line 24: “no obvious oxide at the clad interface the carbon steel and Ni interlayer” →“no obvious oxide at the clad interface between the carbon steel and Ni interlayer”.

(18) Page 6, line 11: “The area corresponding to the point H represents the representative element Cr of stainless steel, and its content reaches 12.15at.%” → “The point H is locatd in the stainless steel cladding, because the content of Cr element reaches 12.15at. %”.

(19) Page 6, line 17: “it can be observed that the grain symbiosis has occurred at the interfaces” → “it can be observed that many grains across the interface transition zone.”

(20) Page 6, line 22: “Its microstructure is same as that of the matrix (Q235), which proves that there is already near the interface and there is absence of decarburized layer.” → “The ferrite and pearlite mixture microstructure near the  are same as that of the matrix (Q235), which proves that there is absence of decarburized layer.”

(21) Page 6, line 26: “there are many chromium carbide” → “there are many chromium carbides”.

(22) Page 7, line 9: “Moreover, there are traces of Fe diffusion in Ni interlayer”.

(23) Page 7, line 16: “Fig. 5(d) shows the stainless steel cladding area far from the interface that many twins” → “Fig. 5(d) shows the austenite stainless steel cladding area far from the interface contains many twins”.

(24) Page 8, line 2: “indicating that the heat treatment has significant solid solution effect on eliminating the sensitization phenomenon.”

(25) Page 8, line 4: “Fig. 5(e) shows the area of the stainless steel layer close to the interface, that a large number of Cr23C6 particles” → “Fig. 5(e) shows the area of the stainless steel layer close to the interface where there are a large number of Cr23C6 particles”.

(26) Page 9, line 17: “the difference of indentation sizes in different regions is obviously decreased, as shown in Fig. 6(b). Moreover, the decarburized layer is disappeared, and the hardness of carburized layer decreases. It can improve the weak area’s strength and crack propagation resistance of the clad plate.”

(27) Page 10, line 15: “That is, as the severe diffusion of elements at high temperature will obviously improve the interface bonding strength.”

(28) Page 12, line 11: “the influence of phase transitions would generated large residual stresses at interface during the quenching, and greatly reduced the shear strength” → “the phase transitions would generate severe residual stresses at interface during the quenching temperature, so that greatly reduced the shear strength”.

(29) Page 13, line 3: “Meanwhile, the intergranular chromium carbide of the carburized layer would be solid-solved at high temperatures. It slows down the generation of intergranular corrosion cracks and effectively improving the plastic deformation capacity of stainless steel cladding.”

(30) Page 14, line 26: “There were obvious carburized and decarburized layers on both sides of the interface due to the excessive diffusion of carbon during the hot rolling process as shown in Fig. 11(a).”

(31) Page 14, line 39: “Obviously, the stainless steel cladding that far from the interface had eliminated the sensitization phenomenon” → “Obviously, there is no sensitization phenomenon in the stainless steel layer that far from the interface”.

(32) Page 15, line 3: “The stainless steel cladding had abnormal growth of crystal grains, leading to the decrease of comprehensive mechanical properties.”

Point 3: It is not simple to understand at which sides stainless and carbon steels are at the fig 6.

Response 3: We have corrected the Fig. 6. Page 10, line 1:

Figure 6. The hardness of stainless steel clad plates. (ac) hardness indentation with different thicknesses of Ni interlayer (Partial area); (d) the distribution of hardness.

Point 4: “In order to ensure the accuracy of the experimental results and ensure a reasonable distance between the indentations” What was done to ensure that?

Response 4: Thanks for the reviewer’s suggestion. In the direction perpendicular to the interface, select an area for hardness test at intervals of 10μm. In this area, perform 5 tests along the parallel interface direction at intervals of 50μm, and take the average value. Page 9, line 7. It can ensure the accuracy of the experimental results and ensure a reasonable distance between the indentations.

Reviewer 2 Report

The article entitled “The effect of Ni Interlayer on the Hot-Rolled and Quenched Stainless Steel Clad Plate” is focused on the changes introduced by the Ni interlayer on the microstructure of the stainless steel clad plate and their mechanical properties.

The article is written in a clear form, all required elements have been included. The title and abstract are consistent and clearly outline the scientific problem. The experiment details together with detailed analysis are well described. The results are presented along with discussion; separately for each experiment and later summarized in the form of a diagram of bonding interface formation. The studies have been precisely summarized in the conclusion section.

Presented in the manuscript work is sufficiently original, I recommend the manuscript for publication.

Small improvements could be provided:

  • Fig 2. a, b, c – the description on those figures is not well visible: especially “carbon steel”, “carbuized layer”
  • I would recommend adding the name of the company/country of used equipment.
  • Spaces between given values and units – sometimes space is used, sometimes it is not
  • The references are presented only in the introduction section – there is no discussion of the results with the literature data. Authors mentioned in the introduction section that systematic studies of studied by them properties/systems were not systematically conducted but some data (even authors' previous work) that address several mentioned in this manuscript issues are available.

Author Response

Response to Reviewer Comments

Point 1: Fig 2. a, b, c – the description on those figures is not well visible: especially “carbon steel”, “carbuized layer”.

Response 1: Thanks for the reviewer’s suggestion. We have corrected the Fig. 2, and it is visible. Page 5, line 1:

Figure 2. The optical microstructure of stainless steel clad plate with Ni interlayers with the thickness of 0, 100 and 200μm and quenching temperatures at (ac) hot rolling; (df) 1000°C; (gi) 1050°C; (jl) 1100°C; (mo) 1150°C.

Point 2: I would recommend adding the name of the company/country of used equipment.

Response 2: Thanks for the reviewer’s suggestion, we have added the name of the company/country of used equipment. Page 3, line 7: “The microstructure and morphology of stainless steel clad plate were observed by optical microscope (OM, Leica, Germany).” “In addition, depth of field optical microscope (Zeiss, Germany) JSM-7100F scanning electron microscope (SEM, JEOL. Japan), Tecnai G2 F30 transmission electron microscope (TEM, FEI, America) and JXA-8530F electron probe microanalysis (EPMA, JEOL. Japan) were used in the experiment to investigate interface microstructure.” “Strength properties were measured at a constant speed of 2mm/min using an AGS-50kNX universal testing machine (Shimadzu, Japan).” “During the HV hardness testing (Shimadzu, Japan), a 100gf test load was applied for all the samples.”

Point 3: Spaces between given values and units – sometimes space is used, sometimes it is not.

Response 3: Thanks for the reviewer’s suggestion, we have modified the format. The format is unified with no spaces. Page 2, line 40; Page 3, line 4; Page 6, line 9; Page 10, line 24.

Point 4: The references are presented only in the introduction section – there is no discussion of the results with the literature data. Authors mentioned in the introduction section that systematic studies of studied by them properties/systems were not systematically conducted but some data (even authors' previous work) that address several mentioned in this manuscript issues are available.

Response 4: Thanks for the reviewer’s suggestion, we have revised the references cited.

(1) Page 2, line 13: “Wang et al. [20] found that the clad plate (SUS316/Q235) fabricated with Ni interlayer possesses the high shear strength, tensile strength, and tensile fracture ductility” → “Wang et al. [20] found that the clad plate (SUS316/Q235) fabricated with Ni interlayer possesses the high shear strength, tensile strength, and tensile fracture elongation, which are 331MPa, 567MPa and 49.63%, respectively. These mechanical properties are high than clad plate without interlayer and Nb or Fe interlayer.”

(2) Page 2, line 17: “Liu et al. [21] reported that the reasonable heat treatment procedure can achieve a high strength-toughness balance of clad plate” → “Liu et al. [21] reported that the intergranular-like corrosion cracks at the carburized layer can be effectively inhibited with the quenching temperature of 1050oC.”.

(3) Page 6, line 18: It indicates that obvious metallurgical bonding has occurred at the clad interface, which will greatly improve the interface bonding strength [20].

(4) Page 9, line 4: carburized layer < stainless steel cladding < carbon steel substrate < decarburized layer, which is in agreement with HV-distance curves in Fig. 6(d) [26].

[20] Wang, S.; Liu, B.X.; Zhang, X.; Chen, C.X.; Fang, W.; Ji, P.J.; Feng, J.H.; Jiang, Y.F.; Yin, F.X. Microstructure and interface fracture characteristics of hot rolled stainless steel clad plates by adding different interlayers. Steel. Res. Int. 2020, 91, 1-14. https://doi.org/10.1002/srin.201900604.

[21] Liu, B.X.; Wang, S.; Fang, W.; Ma, J.L.; Yin, F.X.; He, J.N.; Feng, J.H.; Chen, C.X. Microstructure and mechanical properties of hot rolled stainless steel clad plate by heat treatment. Mater. Chem. Phys. 2018, 216, 460-467. https://doi.org/10.1016/j.matchemphys.2018.06.033.

[26] Chen, C.X.; Liu, M.Y.; Liu, B.X.; Yin, F.X.; Dong, Y.C.; Zhang, X.; Zhang, F.Y.; Zhang Y.G. Tensile shear sample design and interfacial shear strength of stainless steel clad plate. Fusion. Eng. Des. 2017, 125, 431-441. http://dx.doi.org/10.1016/j.fusengdes.2017.05.136.

Reviewer 3 Report

After carefully proofreading, this article is based on the scientific methods for analysis with the properties enhancements with the Ni interlayer adding in SS314/Q235 stainless steel. The minor problems are some typos and too long senteces. The followings are some examples:

In abstract, the third line: "In this study, the comprehensive properties was improved...." Were should be instead of was.

In results and discussion, "Fig. 5 shows the microstructure diagram of ....., which proves that the existence of th Ni interlayer interface bonding state." It's too long sentece could be shorten.

EDS images should be removed to high resolution ones.

Generally, this article is sound and solid scientific results and good arrangement for publication. I am agree it to publish after minor revision.

Author Response

Response to Reviewer Comments

Point 1: In abstract, the third line: "In this study, the comprehensive properties was improved...." Were should be instead of was.

Response 1: Page 1, line 13: “In this study, the comprehensive properties was improved....” → “In this study, the comprehensive properties were improved....”.

Point 2: In results and discussion, "Fig. 5 shows the microstructure diagram of ....., which proves that the existence of th Ni interlayer interface bonding state." It's too long sentece could be shorten.

Response 2: Page 7, line 6: “Fig. 5 shows the microstructure diagram of ....., which proves that the existence of the Ni interlayer interface bonding state.” → “Fig. 5 shows the microstructure diagram of the interface transition zone of a stainless steel clad plate with the addition of 100μm Ni interlayer and quenched at 1050oC. As shown in Figs. 5(a), (b) and (f)-(h), similar to the hot-rolled interface, there is no obvious clad interface between the Ni interlayer and the stainless steel/carbon steel. Moreover, there are traces of Fe diffusion in Ni interlayer, which proves that the existence of the Ni interlayer can effectively improve the interface bonding state.”

Point 3: EDS images should be removed to high resolution ones.

Response 3: Thanks for the reviewer’s suggestion, we have corrected the Figs. 4, 5. The clarity of the picture has been improved. Page 7, line 2; Page 8, line 12:

Figure 4. The TEM microstructure of hot rolled clad plate with 100μm Ni interlayer at interface zone. (a) Ni-Carbon steel; (b) Ni-Stainless steel; (c) Carbon steel; (d) Stainless steel; (e–h) EDS results of interface zone.

Figure 5. The TEM microstructure of 1050oC heat treatment clad plate with 100μm Ni interlayer at interface zone. (a) Ni-Carbon steel; (b) Ni-Stainless steel; (c) Carbon steel; (d) Stainless steel; (e-g) EDS results of interface zone; (h) SAED analysis of (d).
